# Levels and Predictors of Proactive Practical Experience to Solve COVID-19 among Public Health Officers in Primary Care Units in the Upper Southern Region, Thailand: An Explanatory Mixed Methods Approach

**DOI:** 10.3390/ijerph20156487

**Published:** 2023-07-31

**Authors:** Suttida Sangpoom, Femi Adesina, Chuthamat Kaewchandee, Kannika Sikanna, Natchima Noppapak, Sarunya Maneerattanasak, Shamarina Shohaimi, Charuai Suwanbamrung

**Affiliations:** 1Excellent Center for Dengue and Community Public Health: EC for DACH, School of Science, Walailak University, Thasala District, Nakhon Si Thammarat 80160, Thailand; suttida.sa@mail.wu.ac.th; 2Department of Biology, Federal University of Technology Akure, Akura 340110, Nigeria; femi.adesina@outlook.com; 3Community Health Program, Faculty of Liberal Art and Science, Sisaket Rajabhat University, Muang 33000, Thailand; chuthamat.k@sskru.ac.th; 4School of Public Health, Walailak University, Nakhon Si Thammarat 80160, Thailand; kannika.si@st.wu.ac.th (K.S.); natchima.no@mail.wu.ac.th (N.N.); 5Department of Immunology, Faculty of Medicine Siriraj Hospital, Mahidol University, Bangkok 10700, Thailand; sarunya.man@gmail.com; 6Department of Biology, Faculty of Science, Universiti Putra Malaysia, Seri Kembangan 43400, Malaysia; shamarina@upm.edu.my; 7Master and PhD. Program in Public Health Research, Excellent Center for Dengue and Community Public Health: EC for DACH, School of Public Health, Walailak University, Nakhon Si Thammarat 80160, Thailand

**Keywords:** public health officer, COVID-19 solution, mixed methods approach, experience, primary care unit, Thailand

## Abstract

Public Health Officers (PHOs)’ experiences in reaction to the COVID-19 pandemic can be based on whether the PHO is active or passive regarding five experience aspects, including knowledge, understanding, opinion, participation, and practice. Therefore, this study’s objectives are to identify the types of experiences and analyse the predictors of proactive practical experiences in addressing the COVID-19 pandemic among PHOs in the southern region of Thailand. Methods: An explanatory mixed methods approach was used to collect data, through questionnaires and online in-depth interviews. This study was conducted from 4 August 2020 to 3 August 2021. Results: The results include 60 PHOs from 60 Primary Care Units in six provinces, with 41 (68.3%) females and an average age of 35.57 years (SD = 11.61). The PHOs’ knowledge, understanding, and participation experience aspects were mostly proactive rather than passive. The factors that significantly predicted proactive practical experiences included sex (ORadj = 1.52, 95% CI = 1.04–2.21), age (ORadj = 1.69, 95% CI = 1.16–2.48), married status (ORadj = 1.69, 95% CI = 1.16–2.48), education level (ORadj = 1.50, 95% CI = 1.02–2.20), and position for work (ORadj = 1.69, 95% CI = 1.16–2.48). The results of quantitative method were confirmed by 12 sub-themes of 8 PHOs’ experiences from qualitative method. Conclusions: The PHOs’ knowledge, understanding, opinion, and participation experiences were significant predictors of practical experience. Primary health care systems should promote proactive experiences in all four aspects to increase proactive practical experiences.

## 1. Introduction

The COVID-19 pandemic has been a significant concern for healthcare providers worldwide, including Thailand. The Thai healthcare system seeks to rapidly identify and isolate infected patients, provide necessary care, and conduct contact tracing to prevent further transmission [1,2]. In addition, emphasis has been placed on efforts to prevent hospital infections and encourage people to take preventative measures, such as practising hand hygiene, wearing face masks, and getting tested in the case of symptoms. Different provinces have implemented context-specific preventive measures while adhering to national guidelines [3,4]. Due to the work of the public health officers (PHOs), the number of COVID-19 cases in Thailand dropped from a high of 188 recorded in a day (22 March 2020) to a low of 15 on 22 April 2020 [5]. This shows how important the PHO’s job is.

During the pandemic in Thailand, the work of the PHO was important to the COVID-19 management in the primary care unit (PCU). PCUs are a component of a vast network of additional healthcare facilities and are the smallest units that are available all around Thailand. The Ministry of Public Health (MOPH) is responsible for PCUs, which are staffed by PHOs who hold a minimum of bachelor’s degree in health promotion and disease prevention to deliver basic treatment services at the community level. The PHO, which is in charge of screening infected people, assessing risks, investigating diseases, putting in place measures for preventing and controlling infections, spreading information about risks, and making people more aware of the problem, is seen as the first line of defence against the problem. They followed and kept an eye on the rules set out by the Ministry of Public Health to stop COVID-19. This included making sure people refrain from going out, stay away from crowds, keep social distance, work from home, and wear masks when they leave the house. People returning from high-risk locations, both from abroad and from Bangkok and other provinces, were subjected to a 14-day quarantine at home, as well as efforts to prevent the virus from being imported from overseas [6,7].

Following the activities associated with the PHO’s duty mentioned above, this represents their COVID-19 experience. Roth and Jornet [8] define experience as the integration of a person’s knowledge, understanding, opinions, participation, and skills acquired through their learning process in response to events they have encountered at a particular time. This experience can be classified as either passive or active [9]. Passive experience involves gaining knowledge and understanding from reading, hearing, and visualising information from various media, whereas active experience involves the ability to define, plan, or present real-world actions. Therefore, PHOs’ experiences in their PCU in reaction to the COVID-19 pandemic can be broken down into five categories: knowledge, understanding, opinion, participation, and practise. These categories are based on whether the PHO was passive or active. Activities like receiving orders from the district public health office, listening to the COVID-19 situation report, watching announcements from different media, using incidence and prevalence rates on COVID-19, campaigning to educate and promote disease prevention measures, telling people about COVID-19 outbreaks in the community, making pamphlets to give out to people, putting up posters in the community, screening communities for COVID-19, investigating the disease, and reporting to the district public health office were all considered as being proactive [1,6,7]. However, government rules to address COVID-19 are an example of a passive experience [10], such as setting up an emergency operations centre, and mobilise healthcare workers, mostly nurses and public health officers, to enforce 14 days of quarantine and to help collect nasal swabs from all Thai and non-Thai visitors at entry points.

In the past, there was no research pertaining to this level of experience and related factors among PHOs in Thailand. This research is distinguished by a model of experience evaluation that emphasises both quantitative and qualitative aspects. By employing an explanatory mixed methods study, this research aims to evaluate the level of experience and analyse the factor associated to proactive practical experience in addressing the COVID-19 pandemic among PHOs in PCUs from the six provinces in Thailand’s upper southern area. The findings will help the primary healthcare system adjust the determined measures, operational guidelines, and recommendations for resolving future outbreaks.

## 2. Materials and Methods

This research used an explanatory mixed methods design, with the first part being a quantitative method using questionnaires and the second part being qualitative method using the phenomenological study approach through in-depth interviews. Due to the fact that a survey questionnaire will only contain a small number of structured questions, this study explores an explanatory mixed methods approach. Adding qualitative methods may record additional, unforeseen aspects of the topic that may shed more light on solving the research questions and aid in the comprehension of the quantitative data. It can also provide a fuller picture that can improve the descriptions and understanding of the PHO. The data were collected from public health officials (PHOs) who had experience solving COVID-19 problems among public health officers in the upper southern region of Thailand using purposive sampling. This study was conducted after obtaining approval from the Human Research Ethics Committee of Walailak University (No. WUEC-20-218-01), from 4 August 2020 to 3 August 2021.

### 2.1. Quantitative Method

#### 2.1.1. Population and Sample

Based on the experience concept, this study needs be based on provinces with an incidence of COVID-19 patients. At the moment of the research, Thailand’s upper southern area consisted of six provinces dispersed among 23 districts and 302 primary care units. The sample size of the PHOs was determined using the G*Power 3.1 calculus program (http://www.gpower.hhu.de/en.html. 3.11.61, accessed on 10 July 2020) [11], with a test family of “exact” and a statistical test of “bivariate normal model,” with a significance level (α) of 0.05 and a power of 0.95. The program suggested a sample size of 60 PHOs. Stratified sampling was performed at the province and district levels, based on the sample size, followed by simple random sampling to select one PHO from each PCU using a purposive technique (Figure 1).

#### 2.1.2. Instruments of Quantitative Method

This study employed a quantitative design using questionnaires that demonstrated content validity (CVI = 0.96) and reliability (Cronbach’s alpha = 0.83). The questionnaire was developed from a literature review and had six sections. Apart from participant characteristics, knowledge, understanding, opinion, participation, and practical experience were measured using a Likert scale with five levels. Each answer related to knowledge and understanding experience was scored as correct (0 point) and wrong (1 point). We classified the experiences into proactive and passive according to Bloom’s cut-off point of 90% [12]. Proactive knowledge and understanding experience corresponded to a percentage of correct answers ≥ 90% (≥9 points). In contrast, passive knowledge and understanding experience corresponded to <90% (<9 points). Meanwhile, the proactive opinion, practical, and participation experience corresponded to a percentage of correct answers ≥90% (≥45 points), while passive opinion, practical, and participation experience corresponded to a percentage < 90% (<45 points).

#### 2.1.3. Data Analysis of Quantitative Method

The representative characteristics of the 60 participants were analysed for frequency, percentage, mean, and standard deviation. The types of experiences were analysed using a frequency percentage analysis of scores for each aspect of the experience. The factors associated with the practical experiences in solving COVID-19 problems among public health officers were analysed by univariate and multivariate statistics at the 95% confidence interval (95% CI) and at a *p*-value significance < 0.05.

### 2.2. Qualitative Method

#### 2.2.1. Study Design and Participation

An in-depth interview was conducted as part of a phenomenological study to explore how individuals experience certain phenomena in their close environment and derive meaning from them. It is crucial to remember that each person’s phenomenology is distinctly dependent on their experiences [13]. Therefore, the experiences of public health officers involved with COVID-19 solutions may differ and require a tailored approach.

#### 2.2.2. Participant as the Key Informant

A total of eight public health officers from six provinces in the upper southern area of Thailand qualified for the in-depth interviews. This study utilised purposive sampling techniques, including inclusion criteria such as public health officers actively involved in district roles and willing to participate in the interviews. Exclusion criteria were applied, and any public health officer who felt uncomfortable during the interviews was allowed to discontinue.

#### 2.2.3. Guide of Questionnaire for In-Depth Interviews

Three experts validated the unstructured questionnaires used in the interviews, and their results show a content validity index (CVI) value of 0.97. Due to the nature of the public health officers’ roles and responsibilities in preventing and controlling COVID-19 in their districts, this study divided the guide of open-ended questions into six main questions. These included: (1) What do you know about COVID-19 and what are your thoughts about it? (2) Can you share your experiences regarding COVID-19 in your area of responsibility? (3) How have you addressed the COVID-19 situation in the areas you are responsible for? (4) Can you explain how you report on COVID-19 solutions? (5) What has been the response of the community you serve to COVID-19 solutions? and (6) What are the problems, obstacles, and suggestions for solving the COVID-19 problem that you have encountered?

#### 2.2.4. Data Collection of in-Depth Interviews

The research team informed the participants about the objectives of the study, obtained oral consent, and facilitated the public health officers’ views and experiences on COVID-19 solutions. Due to the pandemic situation, the researchers conducted interviews through telephone, line application, and face-to-face, depending on the participants’ availability. The telephone was used to initiate contact with the participants, and interviews lasted range from 30 to 45 min per call to prevent participant fatigue on extended calls. However, each participant had multiple interviews to address the research questions. The major challenges faced were changes in the scheduled time due to the unavailability of the participant and unsaturated responses. If the data were unsaturated, the researchers set the next interview time according to the participants’ condition. For face-to-face interviews, the researchers followed COVID-19 prevention guidelines. The researchers used unstructured questionnaires for the interviews and recorded the sound and tone of the public health officers’ responses.

#### 2.2.5. Data Analysis of Qualitative Method

Various data analysis methods have been utilised in phenomenology research. In this study, thematic analysis was employed, with the five steps outlined by Braun and Clarke [14,15,16]. In each interview, the researcher interviewed the participants, analysed their responses within 24 h, and made note of unclear points to be revisited with the same or next participant.

#### 2.2.6. Trustworthiness

At the end of the qualitative data analysis, the researcher presented the results of the theme to the same eight public health officers so they could thematically review the interpretation of their response and validate it. This technique is the most important one to establish a study’s credibility [17].

## 3. Results

### 3.1. Characteristics Information in Quantitative Method

The results from the self-administered questionnaire about personal data of district public health officers came from 60 public health officers (60 primary care units (PCUs)), who were distributed in six provinces of the upper southern region of Thailand. The participant characteristics comprised almost 41 (68.3%) females, an average age of more than or equal to 27 years for 40 of them (66.7%) (mean = 35.57; SD = 11.61 and range 22–59), a higher education of at least a bachelor’s degree 48 (86.7%), and a public health scholar position 38 (63.3%). Half (30 (50.0%)) of the PHOs had been in their current service position for < or ≥6 years (mean = 12.32, SD = 12.02, and range 1–38 years), and there were 11 (35.0%) COVID-19 patient cases in the district (Table 1).

Among the PHOs surveyed, the top 4 COVID-19 solution activities (Figure 2) with a score higher than 90% were as follows: (1) screening for COVID-19 in the community and conducting disease investigations (100.0%); (2) conducting campaigns to educate and promote disease prevention measures (95.0%); (3) communicating news of COVID-19 outbreaks in the community (93.2%); and (4) receiving orders from the district public health office (91.7%). Additionally, the following activities were also frequently performed: viewing announcements from various media, collecting information and submitting reports to the district public health office, creating pamphlets and posters for distribution in the community, using incidence and prevalence rates for COVID-19 as a reference, and improving environmental sanitation to prevent the spread of COVID-19 with a score below 90%.

### 3.2. Level of PHOs’ Experience to Solve COVID-19 in Upper Southern Region, Thailand

Out of the 60 public health officers surveyed, 34 (56.7%) had categorised experiences that were at a proactive level, while 26 (43.3%) had experiences that were at a passive level. In terms of the specific experience aspects, their experiences in knowledge, understanding, and participation were almost exclusively at a proactive level, with frequencies (percentages) of 48 (80.0%), 38 (63.3%), and 32 (53.3%), respectively. However, their experiences of opinion were predominantly at a passive level, with a frequency (percentage) of 37 (61.7%). Furthermore, the experiences of practicality were equally distributed between proactive and passive levels, with frequencies (percentages) of 30 (50%) and 30 (50.0%), respectively (Figure 3).

### 3.3. Predictor of the Proactive Practical Experiences to Solve COVID-19 among PHOs in Upper Southern Region, Thailand

A study conducted among public health officers revealed that various factors were associated with having proactive practical experiences in addressing COVID-19 (Table 2). Male officers were found to have 1.52 times greater odds of having practical experiences compared to female officers (ORadj = 1.52, 95% CI = 1.04–2.21), and officers under the age of 27 had 1.69 times greater odds of having practical experiences compared to those who were 27 or older (ORadj = 1.69, 95% CI = 1.16–2.48). Single, divorced, or separated officers had 1.69 times greater odds of having practical experiences compared to married officers (ORadj = 1.69, 95% CI = 1.16–2.48), and officers with a bachelor’s degree had 1.50 times greater odds of having practical experiences compared to those with a master’s degree (ORadj = 1.50, 95% CI = 1.02–2.20). Public health practitioners had 1.69 times greater odds of having proactive practical experiences compared to public health scholars (ORadj = 1.69, 95% CI = 1.26–2.48). Additionally, PHOs with proactive knowledge, understanding, opinion, and participation experiences had significantly greater odds of having proactive practical experiences in solving COVID-19.

### 3.4. The Theme of PHOs’ Experiences in Solving COVID-19 from a Qualitative Method

The eight public health officers from the upper southern region of Thailand who volunteered to participate comprised four females and four males, two of whom were single and six of whom were married. In terms of education, five held bachelor’s degrees and three held master’s degrees. The age range of the participants was from 24 to 60 years old, with a mean age of 41.13 years (SD = 11.44). The participants had a range of experience in public health service, with the duration of service ranging from 1 to 40 years and a mean of 15.75 years (SD = 13.13). The number of patients under their care varied from 0 to 65, with a mean of 10 (SD = 22.60). This study identified 12 sub-themes from the questions asked related to the experiences of these public health officers in dealing with COVID-19. These are presented in Table 3.

## 4. Discussion

At the time of this investigation, there were eleven COVID-19 patient cases in the district. Meanwhile, as of 17 July 2020, 744 COVID-19 cases had been confirmed throughout southern Thailand [18,19], which was attributed to individuals from a cluster of COVID-19 patients in Iran and Malaysia travelling to the southern provinces of Thailand, Nakhon Si Thammarat and Surat Thani, in March 2020, and potentially exposed to a number of people. Between 19 and 25 January 2022, the highest rates of new COVID-19 cases were still identified in the southern tourist hotspots of Phuket and Phangnga [2]. However, Thailand’s primary health care (PHC) infrastructure played a significant role in preventing the spread of the COVID-19 virus. Today, the PHC has a highly developed network that connects to each sub-district via primary care unit (PCU) and then to the household level via a network of village health volunteers (VHVs) [20,21].

Among the COVID-19 solution activities mentioned, more than 90% of the PHOs in this survey believe that screening for COVID-19, education campaigns, sharing outbreak news, and adherence to district order are the most effective. In support, Ogboghodo et al. [22] stated that screening at the crucial interface and other facility ports of entry is a must for the rapid identification and isolation of infected patients during outbreaks of highly infectious diseases such as the COVID-19 pandemic. Similarly, Grey et al. [23] mentioned that health and hygiene initiatives, encouraging individuals to change their behaviours, and reinforcing consistent messages are successful ways of lowering infection rates. It is crucial that communities recognise the virus as a health danger and follow the public health office’s orders for preventive actions to combat it. There will be widespread and terrible effects if people are unwilling to do this.

In the southern part of Thailand, almost all of the PHOs’ experiences with COVID-19 in terms of knowledge, understanding, and participation were at a proactive level. According to Geertshuis et al. [24], individuals with high proactive behaviour actively seek out and take advantage of many chances, show initiative, take action, and endure until their goals are met. They are driven and committed to making a difference in the lives of those around them. Health workers should be proactive agents of change whose knowledge protects populations from viral pandemics. Knowledge of health workers is crucial for efficient infection protection and control (IPC). Geberemariyam et al. [25] and Assefa et al. [26] found that a lack of knowledge of infection protection and control guidelines, as well as a lack of awareness of preventive indications during daily patient care and the potential risks of microorganism transmission to patients, are obstacles to solution activities. This will result in delayed diagnosis of new cases, infection dissemination, and inadequate infection control practises. Therefore, PHOs must be well-informed about the pathogen and disease in order to wage an effective war against the virus [27].

In addition, Cheng et al. [28] recognise the critical relevance of infection control readiness in the healthcare system based on their previous experiences with new respiratory illnesses. Subramaniam et al. [29] also said that the fact that workers do not take part in activities to find solutions is a big reason why people do not follow the rules. In this study, a significant number of PHOs took part in solution activities by attending COVID-19-related network meetings, where they talked about ways to teach the public about COVID-19 and got the general public involved in bringing attention to COVID-19 guidelines. Likewise, health workers were proven to be effective in emergency management during the Ebola outbreak in Guinea, Sierra Leone, and Liberia by rapidly identifying cases, as well as in pandemic preparedness by effectively communicating in a culturally relevant manner, educating and mobilising communities, contributing to surveillance systems, and bridging gaps in health service provision.

However, the majority of their experiences with opinion were passive. More than half of PHOs believe the disease to be terrifying since, because it cannot be controlled or prevented, it can result in death, and it poses a risk to their line of work. This is in line with what Ioannou et al. [30] found, that a large number of healthcare workers (HCWs) are also afraid of contracting the virus and passing it to their families. This is a very important finding because anxiety and fear of contracting COVID-19 may affect the quality of care given and may make health officers less willing to take part in solution activities. In a study conducted in Lebanon, 24% and 23% of health workers were reported to have anxiety and depression, respectively, due to the belief that their job put them at great risk, fear of falling ill and dying, concerns about passing COVID-19 onto family and friends, other people avoiding healthcare workers’ families due to their work, and so on [31]. This anxiety level was later reduced after the lockdown, as the majority of healthcare workers showed mild to minimal anxiety during the pandemic. The researchers found that this was strongly linked to the health workers’ high resilience, which was built by the public’s gratitude for “Lebanese masked heroes”, which could have made them more excited and willing to fight the pandemic [32]. Contrary to this, Olateju et al. [33] found that despite participants’ awareness of the increased risk of COVID-19 exposure due to their role, they were still willing to work because they believed that exposure to diseases was an integral part of their job as healthcare workers. Also, social psychology theories say that altruistic behaviour wins out over negative feelings and anxiety, which makes people more likely to do self-sacrificing things when they are in demanding circumstances [34]. Again, a significant proportion of PHOs in this study are uncertain as to whether or not everyone should participate in COVID-19 prevention. Infection control should not be viewed as the responsibility of a select few individuals; rather, it is the result of a collective effort by all healthcare professionals [35] and the society as a whole.

Furthermore, the experiences of practicality were equally distributed between proactive and passive levels. Many are proactive in conducting on-site inspections to ensure compliance with COVID-19 guidelines, and reporting data to the district public health department, while some are passive in investigating diseases with people at risk of contracting COVID-19 and developing plans to prevent and control COVID-19. In an optimal setting, health professionals should serve as health promoters. Putting a significant amount of effort into encouraging behaviour change by serving as role models, defining goals, and forming effective partnerships with stakeholders. By engaging in these activities, they help to strengthen connections between society and the healthcare system and reduce compliance dropouts [36].

The findings of this study conducted among public health officers revealed that gender, age, marital status, education level, and work position were associated with having proactive practical experiences in addressing COVID-19. For instance, male officers were found to have 1.52 times greater odds of having practical experience compared to female officers. This may be because most of the PHOs that were studied were female and many of them were married. Women reported being more likely than males to spend more time caring for children and likely handle the home obligations during the pandemic, which may reduce their practical experience [37]. Contradictions between work and motherhood are further illuminated by the maternal body, which is itself perceived as an inconvenience in professional settings, especially during pregnancy [38]. In contrast, the majority of participants in a study by Regenold et al. [39] opined that their gender had no impact on their experiences as healthcare workers during COVID-19, while the researchers’ findings show otherwise. However, their participants also stated that people’s experiences during the COVID-19 pandemic were based on the roles they played, even though these roles are based on gender. The work title has taken the place of gender, but the same gendered assumptions are still present in job descriptions and affect how diverse jobs are valued, supported, and granted authority. Furthermore, compared to PHOs who were 27 or older, PHOs under 27 had a 1.69 times greater likelihood of having practical experience. Murman [40] claims that there are detectable declines in cognitive function associated with ageing. The most notable shifts are the decreases in cognitive functions including processing speed, working memory, and executive cognitive function that are necessary for making rapid, well-considered decisions. However, expertise gained throughout a lifetime is retained well into old age.

We also discovered that single, divorced, or separated PHOs were 1.69 times more likely to have practical experience than married officers. This is consistent with the findings of the Regenold et al. [39] study, in which a significant number of healthcare employees claimed that mothers and fathers are put in an especially difficult situation when balancing work and home obligations. As previously stated, Mele et al. [37] found that healthcare workers who identified as female and had dependents under the age of ten experienced the highest levels of anxiety during the pandemic. The married PHOs’ household obligations and fear about infecting their family with COVID-19 may have limited their practical experience.

Accordingly, the eight PHOs interviewed to discuss the topic expressed their feelings. Many perceived COVID-19 to be a severe outbreak due to the lack of a specific treatment and its rapid spread. One PHO attributed the rapid spread of the infection in the southern district of Thailand to the reunification of international travellers with their families without screening. Amzat et al. [41], from Nigeria, also affirmed that the majority of those who entered the country did not adhere to the Nigerian Centre for Disease Control’s (NCDC) 14-day self-isolation recommendation. To effectively disseminate information about COVID-19, a line or Facebook network group is utilised to broadcast messages, posters and pamphlets are created and disseminated throughout the community, and public health officers and village health volunteers are involved in correcting practises. As one of the PHOs stated, it is essential to increase community understanding in order to effectively prevent COVID-19. Those who misunderstood the prevention guidelines and did not comprehend quarantine procedures will benefit greatly from this. Some people may mistake the symptoms of COVID-19 for the common cold, particularly if their immune system is powerful enough to mask the virus. The virus’ incubation period can last for several days, and asymptomatic individuals can still be carriers. However, laboratory tests are required to accurately detect the virus, as symptoms may be absent, making prevention challenging [42].

## 5. Conclusions

The findings of this study show that the proactive experience of the PHOs has a significant influence on COVID-19 prevention and control. The participants’ experiences of opinion in this study are predominantly at a passive level, which is inadequate as it has an impact on how committed PHOs will be in delivering quality health care. This may have resulted in passive levels of practical experience that were recorded in this study among the PHOs as well. Through doing and personal experience, knowledge leads to a much deeper comprehension of a concept; therefore, understanding the infection can help PHOs become more practical. This calls for leadership to provide infection prevention and control training and education, as required.

## Figures and Tables

**Figure 1 ijerph-20-06487-f001:**
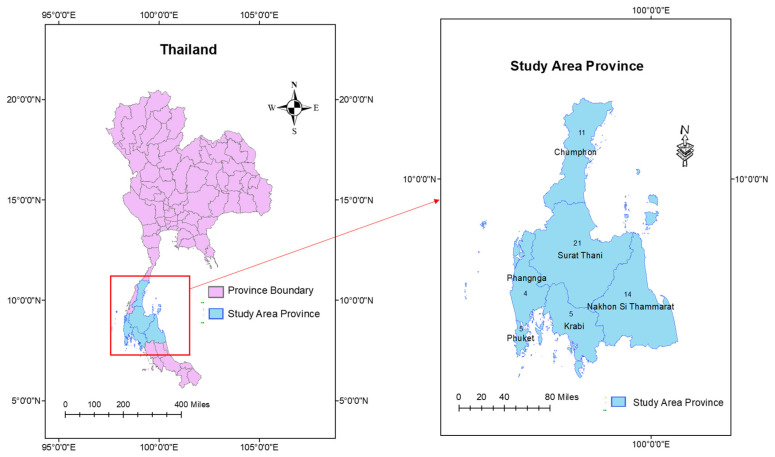
Mapping 60 PCUs in 6 provinces, upper southern region, Thailand.

**Figure 2 ijerph-20-06487-f002:**
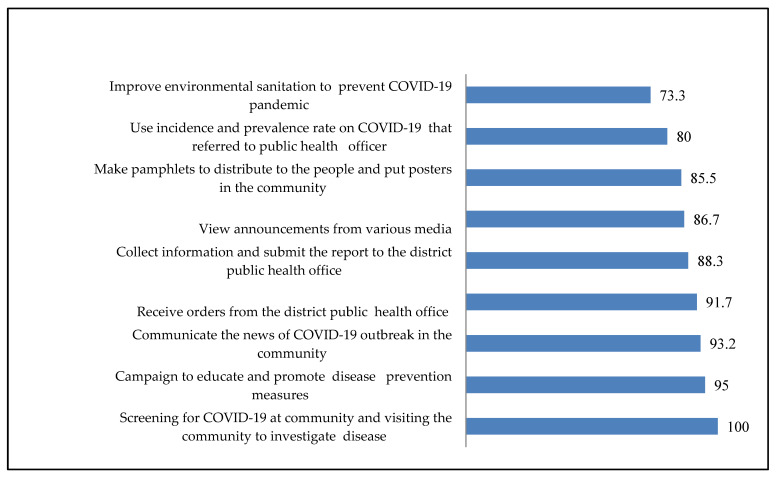
PHOs’ activities to solve COVID-19 in upper southern region, Thailand.

**Figure 3 ijerph-20-06487-f003:**
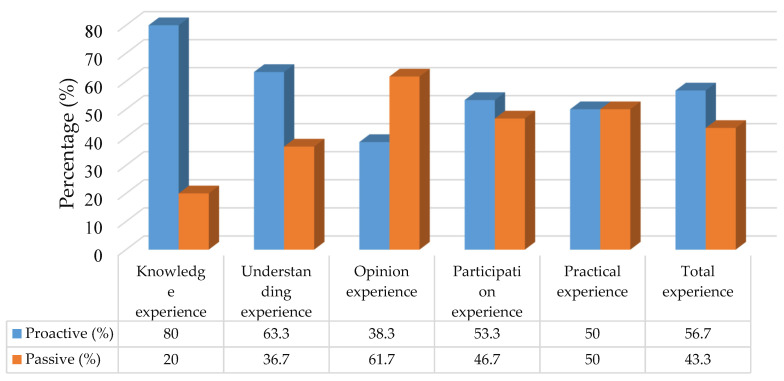
PHOs’experience to solve COVID-19 in upper southern region, Thailand.

**Table 1 ijerph-20-06487-t001:** Characteristics information of the 60 PHOs from 60 PCUs.

Characteristics	n (%)
Sex	
Male	19 (31.7)
Female	41 (68.3)
Age (Year old) $\bar{X}$ (S.D.) = 35.57 (11.61), Min = 22, Max = 59	
<27	20 (33.3)
≥27	40 (66.7)
Marital status	
Single/divorced/separated	29 (48.3)
Married	31 (51.7)
Education level	
Bachelor’s degree	48 (86.7)
Master’s degree	12 (13.3)
Public health position	
Public Health Scholar	38 (63.3)
Public Health Practitioner	22 (36.7)
Length of time worked in the current position $\bar{X}$ (S.D.) = 12.32 (12.02), Min = 1, Max = 38	
≤6 years	30 (50.0)
>6 years	30 (50.0)
COVID-19 patients in the area	
No	49 (75.0)
Yes	11 (25.0)

**Table 2 ijerph-20-06487-t002:** Predictor of proactive practical experiences to solve COVID-19 among PHOs in the upper southern region of Thailand.

Factors	Proactive Practical Experiences (n = 30)	OR ^a^	OR_adj_ ^b^	95% CI	*p*-Value
Sex					
Male	9	0.86	1.52	1.04–2.21	0.029 *
Female	21		1.00		
Age (years old)					
<27	11	1.35	1.69	1.16–2.48	0.006 *
≥27	19		1.00		
Marital status					
Single/divorced/separated	12	0.51	1.69	1.16–2.48	0.006 *
Married	18		1.00		
Education level					
Bachelor’s degree	24	1.00	1.50	1.02–2.20	0.040 *
Master’s degree	6		1.00		
Position for work					
Public Health Practitioner	23	3.28	1.69	1.16–2.48	0.006 *
Public Health Scholar	7		1.00		
Length of time worked in the current position					
>6 years	17	1.71	0.93	0.60–1.46	0.829
≤6 years	13		1.00		
Presence of COVID-19 patients in the area					
No	25	1.25	1.24	0.84–1.82	0.301
Yes	5		1.00		
Knowledge experience					
Proactive	24	1.00	1.65	1.13–2.40	0.008 *
Passive	6		1.00		
Understanding experience					
Proactive	23	3.76	1.47	1.01–2.14	0.045 *
Passive	7		1.00		
Opinion experience					
Proactive	7	1.22	1.48	1.01–2.21	0.046 *
Passive	23		1.00		
Participation experience					
Proactive	24	11.00	1.58	1.08–2.31	<0.017 *
Passive	6		1.00		

* *p*-value < 0.05, Goodness of fit = 0.636, Adjusted for Education level and Position for work; ^a^ Univariate analysis, Chi-squared test, ^b^ Multivariate analysis, Multiple logistic regression.

**Table 3 ijerph-20-06487-t003:** Sub-theme, meaning, and responses from the 8 PHOs’ experiences in upper southern region, Thailand.

Sub-Themes	Meaning	Responses
Perceptions about the pandemic and awareness of COVID-19.	Significant impact on the world. - Spreading rapidly and unexpectedly.	It was a pretty serious outbreak because there was no specific cure and it happened so quickly. In the infected area, there is a spread of COVID-19 infection among those who returned from travel and reunited with people at home (PHO1).
COVID-19 knowledge leads to standard practice and guidelines.	Educating people on preventive measures and control strategies: - Wearing masks. - Frequent hand washing. - Social distance. - Promoting illness awareness. - Dissemination of information on social media. Each community needs the same standard guidelines.	It began with forming a team with a network to create a Facebook and video on the online application to publicise and educate people in various places, such as mosques, schools, government service facilities, and hotels. Distributing leaflets and pasting posters in the community as an education program. This highlights the beginning of the program’s implementation process (PHO5). Preventing and controlling COVID-19 in quarantine areas, promoting mask-wearing, and social distancing in the public (PHO1).
A careful understanding of COVID-19 solutions is necessary.	People must comprehend the public health officer’s methods for preventing COVID-19. - Comprehend prevention guidelines.	During the first phase of the pandemic, people misunderstood the prevention guidelines and did not understand quarantine procedures. They entered the village without notifying their public health officer and bypassed checkpoints set up by primary care units, entering the community through natural channels. It is crucial to enhance people’s understanding in the community to effectively prevent COVID-19 (PHO6).
Recognising the nature and severity of COVID-19.	Understanding the symptoms associated with the virus: - High fever lasting 2–3 days, diarrhoea, difficulty breathing, coughing, wheezing, fatigue, sore throat, and nasal congestion.	As previously mentioned, COVID-19 is a severe disease that can cause damage to the lungs and lead to death. Although death may not occur, individuals may still experience lasting symptoms that affect their daily lives. Therefore, if an infection does occur, it must be taken care of to prevent potential complications (PHO4).
COVID-19 impacts lifestyle and quality of life.	Individuals returning from high-risk areas must be quarantined and separated from their families. Not visiting others while sick.	It has an impact on the way of life of people in the area, with long-term effects on the economy, society, and the livelihoods of people in all areas (PHO5).
Respond to and trust public health officials to solve COVID-19.	To trust and cooperate with public health officials to combat COVID-19.	A successful process involves participation from all sectors. When a disease outbreak occurs, everyone tends to rely solely on public health officials to handle it. However, with the prevalence of COVID-19, all sectors must contribute to prevention and control efforts to improve outcomes. If public health officials are the only ones responsible, controlling the disease becomes challenging. Making everyone a stakeholder in preventing and controlling the disease is crucial (PHO5).
Establishment of a collaborative network of stakeholders to address the COVID-19 pandemic.	The cooperation between various stakeholders, such as government agencies, private organisations, and the local community, to tackle the challenges posed by COVID-19.	The network of communities begins at the district level, including the district public health office, sub-district health centre, village head, and village health volunteer, all of whom participate in the collaborative effort. Coordination with the network of communities involves writing letters to request support and assistance to fill any remaining gaps (PHO 7).
Stakeholders’ contribution to spreading awareness and monitoring COVID-19 treatment.	- Disseminating knowledge about COVID-19 prevention. - Screening individuals with suspected symptoms, etc. - Providing facilities for infected people.	If companies provide alcohol, it can be distributed to schools, community organisations for funeral use, and placed in various locations in the community. It can also serve as a model for villagers. Knowledge is shared by transmitting it to over 130 community organisations, and each responsible community organisation will then pass on the knowledge about COVID-19 prevention (PHO2).
COVID-19 screening and referral checkpoints.	Establishing checkpoints in various locations, such as schools, hotels, temples, mosques, and village extraction points.	If someone in quarantine has a suspected disease, we screen them and send them to the district hospital for investigation. For the common people in the community, there were checkpoints at the screening points in the village. This indicates the importance of taking proactive measures to prevent the spread of COVID-19 in the community (PHO3). I have been involved in every aspect of combating COVID-19, from setting up screening and referral checkpoints in schools, hotels, temples, and mosques to working together to prevent the spread of the disease, providing knowledge, monitoring and taking care of high-risk groups in isolation for 14 days, and searching for infected patients (PHO6).
Collaboration and sharing opinions to improve solutions.	The process of working together and gathering ideas from different communities to find solutions to problems.	We work with network leaders, the local administration organisation head, and the district public health office to ensure adequate supplies. The local administration organisation will provide support for food, and the primary care unit of the public health office will continue to provide COVID-19-related knowledge to the community (PHO8).
Conducting community examinations, follow-ups, and reporting.	The process of implementing an operational plan to monitor and track individuals who may have been exposed to COVID-19 in the community.	The operational plan creates group lines for each group in the community and brings people at risk into the group line to report symptoms for 14 days. If there is no confirmed case, that person will be immediately removed from the group line (PHO1).
Today’s practical prevention and control lead to future solutions.	Proactive plan.	At first, there is a passive plan, but when there is a COVID-19 patient, the plan changes to a proactive plan, such as the meeting of the district committee every week to adjust the plan, setting up additional checkpoints to screen employees. For the second round, there are meetings with all stakeholders, such as the local government and the police department, to solve the problem (PHO5).

## Data Availability

All datasets are available upon request to the corresponding authors.

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
