# Peer review of "Levels and Predictors of Proactive Practical Experience to Solve COVID-19 among Public Health Officers in Primary Care Units in the Upper Southern Region, Thailand: An Explanatory Mixed Methods Approach"

_ijerph, 2023, doi:10.3390/ijerph20156487_

Round 1

Reviewer 1 Report

The manuscript describes a mixed-methods investigation into the role of public health officers in Thailand during the COVID-19 pandemic. The topic is interesting and relevant for the readership of IJERPH.

There are some aspects of the manuscript that need to be improved before the manuscript can be published, though.

Title:

The title contains the term “experience” twice and is rather long and winded. Please find a more concise wording.

Abstract:

The first sentence needs to be reformulated. It is currently not understandable what the terms “passive” and “active” refer to. Please elaborate!

Materials and methods:

While the methods are generally described in an accessible way, I am missing a reflection and legitimation of the mixed-methods approach. I would encourage the authors to argue why such an approach is chosen for the particular context and problem studied. The approach seems well-suited, but the manuscript would profit from briefly discussing this.

Line 134: Please change “interview” to “interviews” – there were several of them.

Line 136: This sentence needs to be better explained or reformulated.

Also, the qualitative approach should be better described – what are challenges related to interviewing people online? How can an interview of 45 min be considered phenomenological? Please clarify these aspects.

Line 163: It is quite challenging to establish a relationship with people over phone. Maybe “initiate contact” would be a more appropriate term?

Findings:

The reporting on the quantitative part seems solid. The data presentation of the qualitative part is naturally more limited in a mixed-methods study than in a purely qualitative study. The chosen format of Table 3 structured according to sub-themes and with meaning rather than interpretation seems adequate for this study.

Discussion:

Line 305-307: Here you discuss anxiety in connection with COVID-19 health risks. This is an important aspect, but you fail to engage with existing literature on the topic. Please see the work of Fersch et. al. (2022) on anxiety and trust in the interaction of citizens with frontline staff in public services published in the journal Health, Risk & Society. Please engage with this work on aspects of anxiety and health risks of frontline staff during the COVID-19 pandemic (where both public health officers and primary school teachers filled important roles).

Also, you discuss aspects of gender and civil status, but refrain from deeper reflections on the matter. Doing so might further ground this manuscript in the existing literature. Please engage with Regenold & Vindrola-Padros (2021) on how gender shaped the experiences of healthcare workers (such as your public health officers) during the COVID-19 pandemic.

Fersch et al. (2022). Anxiety and trust in times of health crisis: How parents navigated health risks during the early phases of the COVID-19 pandemic in Denmark. Health, Risk & Society, Taylor & Francis. https://doi.org/10.1080/13698575.2022.2028743

Regenold & Vindrola-Padros (2021). Gender Matters: A Gender Analysis of Healthcare Workers’ Experiences during the First COVID-19 Pandemic Peak in England. Social Sciences, MDPI. https://doi.org/10.3390/socsci10020043

When integrating these two references in the Discussion, please make sure these (and any supplementary ones) are integrated into the Introduction to the degree necessary.

Author Response

Dear Editor and Reviewers,

Many thanks to the reviewers for his/her comments and suggestions. 

Point 1: The title contains the term “experience” twice and is rather long and winded. Please find a more concise wording.

Response 1: This has been adjusted. We now have "Level and predictors of proactive practical experience to solve COVID-19 among public health officers in primary care units in the upper southern region, Thailand: an explanatory mixed methods approach

Point 2: The first sentence needs to be reformulated. It is currently not understandable what the terms “passive” and “active” refer to. Please elaborate!

Response 2: It is the abstract needs to shortcut. However, the study expained in the topic of tool that PHO’s experience means five experience aspects included knowledge, understanding, opinion, participation, and practice to solve COVID-19. Based on Bloom’s cut-off  point, the study divided score active experience identify cover more ≥90% and passive is lower than 90% of total score.

 In the firsth sentence of the abstract,  we write “Public Health Officers' (PHO) experiences in reaction to the COVID-19 pandemic can be based on whether the PHO is active and passive of five experience aspects included knowledge, understanding, opinion, participation, and practice.”

Point 3: While the methods are generally described in an accessible way, I am missing a reflection and legitimation of the mixed-methods approach. I would encourage the authors to argue why such an approach is chosen for the particular context and problem studied. The approach seems well-suited, but the manuscript would profit from briefly discussing this.

Response 3: It has been discussed. Due to the fact that a survey questionnaire will only contain a small number of structured questions, this study explores an explanatory mixed-methods approach. Adding qualitative methods may record additional, unforeseen aspects of the topic that may shed more light on solving the research questions and aid in the comprehension of the quantitative data.

Point 4: Please change “interview” to “interviews” – there were several of them.

Response 4: It has been changed as appropriate.

Point 5: Also, the qualitative approach should be better described – what are challenges related to interviewing people online? How can an interview of 45 min be considered phenomenological? Please clarify these aspects.

Response 5: Clarification has been made. "The telephone was used to initiate contact with the participants, and interviews lasted between 30-45 minutes per call to prevent participant fatigue on extended calls. However, each participant had multiple interviews to address the research questions. The major challenges faced are the change in the scheduled time due to the unavailability of the participant and unsaturated responses."

Point 6: It is quite challenging to establish a relationship with people over phone. Maybe “initiate contact” would be a more appropriate term?

Response 6:  Changed – “The telephone was used to initiate contact with the participants, and interviews lasted between 30-45 minutes.”

Point 7: Line 305-307: Here you discuss anxiety in connection with COVID-19 health risks. This is an important aspect, but you fail to engage with existing literature on the topic. Please see the work of Fersch et. al. (2022) on anxiety and trust in the interaction of citizens with frontline staff in public services published in the journal Health, Risk & Society. Please engage with this work on aspects of anxiety and health risks of frontline staff during the COVID-19 pandemic (where both public health officers and primary school teachers filled important roles).

Response 7: Done. Although the authors used other suitable papers rather than the suggested paper (Fersch et al., 2022) by the reviewer, this is due to the fact that the suggested paper discussed the anxiety of parents towards their children's school resumption after COVID-19, not specifically healthcare workers’ experiences during COVID-19.

“In a study conducted in Lebanon, 24% and 23% of health workers were reported to have anxiety and depression, respectively, due to the belief that their job put them at great risk, fear of falling ill and dying, concerns about passing COVID-19 on to family and friends, other people avoiding healthcare workers' families because of their work, and so on (Msheik El Khoury et al., 2021). This anxiety level was later reduced after the lockdown, as the majority of healthcare workers showed mild to minimal anxiety during the pandemic. The researchers found that this was strongly linked to the health workers' high resilience, which was built by the public's gratitude for “Lebanese masked heroes”, which could have made them more excited and willing to fight the pandemic (Sakr et al., 2022).”

Point 8: Also, you discuss aspects of gender and civil status, but refrain from deeper reflections on the matter. Doing so might further ground this manuscript in the existing literature. Please engage with Regenold & Vindrola-Padros (2021) on how gender shaped the experiences of healthcare workers (such as your public health officers) during the COVID-19 pandemic.

Response 8: Done. “Contradictions between work and motherhood are further illuminated by the maternal body, which is itself perceived as an inconvenience in professional settings, especially during pregnancy (Hennekam et al., 2019).”

“In contrast, the majority of participants in a study by Regenold et al. [35] opined that their gender had no impact on their experiences as healthcare workers during COVID-19, while the researchers findings show otherwise. Although, their participants also stated that people's experiences during the COVID-19 pandemic were based on their roles played, even though these roles are based on gender. The work title has taken the place of gender, but the same gendered assumptions are still present in job descriptions and affect how diverse jobs are valued, supported, and granted authority.”

Reviewer 2 Report

This paper addresses a topic that has been addressed by many.  My primary concerns relate to a lack of understanding of the healthcare system in Thailand.  First, I would like to see more detail on the relationship between the PHO and the PCU.  For me, a primary care unit is a site that delivers care to patients.  Do PHOs work in a PCU and deliver care to patients? there are times when you seem to use the terms interchangeably.  For example, the caption in Figure 1 uses 60 CPUs.  I think that should be PHOs.  Also, How are PHOs assigned?  Are they assigned to provinces, districts, or villages?  What is the authority of a PHO? 

You distinguish between passive and active.  It seems that learning about Covid is a passive activity although what the person is doing is very active.  My point is that it seems to me to be active, one must first learn about Covid to understand what to do.  I do think if the PHO does nothing passive, my comments relate to the definitions of the terms in lines 60-75.   

Another area is that I would like to have a better description of what the government through the Department of Public Health's role is in providing guidelines and requirements to the PHOs.  I would think that information would be precise, and the PHOs all would know what is required. Why is the behavior left entirely up to each PHO?  

Your study encompasses a large area, covering 7 provinces. Figure 1 says the 60 PHOS are in 6 provinces.  Please clarify.  There must be many differences in the conditions across this large expanse of geography. What impact did the number of cases of COVID-19 in an area have on the actions of the PHO?  I think the questions In lines 151-157 might be interpreted differently by an individual. I think some of your questions are not clear and could be misinterpreted. Number 4 seems very open-ended.  What are the choices?  Are these not prescribed? Number 3 is very general and can be answered in many different ways.  Was Zoom not an option for conducting interviews as a substitute for face-to-face?  Not clear what line application (line 162) means.  What is the difference between a public health scholar and a public health practitioner? Line 185 - how can you have almost 41 females?

The actual interview section was interesting, although 8 is a very small number. I am not sure what conclusions one can make, except the remarks you include seem to be very predictable. What I would be more interested in would be the differences in the remarks. What comments were the same and how many and what and how many were different?

Finally, I find the conclusions very weak.  What were you trying to accomplish by writing this paper?  Was the purpose to get change?  Was it to identify problems that should be corrected?  There is little evaluation of what is good and what is less good.   The Conclusions should be the strongest part of the paper.  What did you intend to be of value to the reader, including readers from another country?

A few typos but basically excellent.

Author Response

Dear Editor and Reviewers,

Many thanks to the reviewers for his/her comments and suggestions. 

Point 1: My primary concerns relate to a lack of understanding of the healthcare system in Thailand.  First, I would like to see more detail on the relationship between the PHO and the PCU.  For me, a primary care unit is a site that delivers care to patients.  Do PHOs work in a PCU and deliver care to patients? there are times when you seem to use the terms interchangeably.  For example, the caption in Figure 1 uses 60 CPUs.  I think that should be PHOs.  Also, How are PHOs assigned?  Are they assigned to provinces, districts, or villages?  What is the authority of a PHO? 

Response 1: About the title in Figure 1, there are 302 PCUs distributed across six provinces in the southern part of Thailand, of which 60 PCUs were selected for this study, and one PHO was recruited from each selected PCU. Ranong Province was not included in this study because there were no COVID-19 patients. That is why the figure is titled "Mapping 60 PCUs in 6 Provinces, Upper Southern Region, Thailand".

Added to the paper – “PCUs are a component of a vast network of additional healthcare facilities and are the smallest units that are available all around Thailand. The Ministry of Public Health (MOPH) is responsible for PCUs, which are staffed by PHOs who hold a minimum of bachelor's degree in health promotion and disease prevention to deliver basic treatment services at the community level.”

PHOs can be assigned to provinces, districts, or villages.

Point 2: You distinguish between passive and active.  It seems that learning about Covid is a passive activity although what the person is doing is very active.  My point is that it seems to me to be active, one must first learn about Covid to understand what to do.  I do think if the PHO does nothing passive, my comments relate to the definitions of the terms in lines 60-75.   

Response 2: PHO that are not engaged in the listed active roles (61-75), will be passive towards disease control. Ideally, PHO should do nothing passive, but really, workers display different attitudes towards their job. Learning about diseases is an active activity.

Point 3: Another area is that I would like to have a better description of what the government through the Department of Public Health's role is in providing guidelines and requirements to the PHOs.  I would think that information would be precise, and the PHOs all would know what is required. Why is the behavior left entirely up to each PHO?  

Response 3: Government guidelines such as the setting up of an emergency operations centre and mobilisation of health care workers, mostly nurses and public health officers, to enforce 14 days of quarantine and to help collect nasal swabs from all Thai and non-Thai visitors at entry points.

Point 4: Was Zoom not an option for conducting interviews as a substitute for face-to-face?  Not clear what line application (line 162) means.

Response 4: We referred to Zoom as an online application, but not all participants have access to Zoom.

Point 5: There must be many differences in the conditions across this large expanse of geography.What impact did the number of cases of COVID-19 in an area have on the actions of the PHO?

Response 5: This was mentioned in the discussion part: "Between January 19 and 25, 2022, the highest rates of new COVID-19 patients were still identified in the southern tourist hotspots of Phuket and Phangnga". Generally, in the majority of the areas where there is a high number of cases, stress and anxiety are associated with the PHOs.

Point 6:  I think the questions In lines 151-157 might be interpreted differently by an individual. I think some of your questions are not clear and could be misinterpreted. Number 4 seems very open-ended. What are the choices?  Are these not prescribed? Number 3 is very general and can be answered in many different ways. 

Response 6: This part of the study used the qualitative method, which uses open-ended questions to collect data. In order to explore the participant’s experience with COVID-19 and not limit their responses, no choices were given to the questions. From the general answers of the participants, sub-themes were written out, which is the ideal of using the phenomenological qualitative method in this study.

Point 7: What is the difference between a public health scholar and a public health practitioner?

Response 7: Thailand's Office of Civil Service Commission recommends regulations for the career portion of health provider in December 2008 (https://www.ocsc.go.th/job/ access 09_07_2023). The role of public health scholar has been defined as a PHO who works as a primary practitioner and is required to work, do work linked to academic work in public health under supervision, suggest, inspect, and perform other activities as assigned. Public health practitioners, on the other hand, are PHO who work as a supervisor and must oversee, advise, and inspect the work of coworkers. Use knowledge, competence, experience, and high expertise in public health academic work to complete assignment. Perform duties that need tough judgments or the resolution of difficulties, as well as other responsibilities as given or as an experienced practitioner by using knowledge, skills, experience and high expertise in public health academic work Perform tasks that require difficult decisions or problems to solve and work other as assigned.

Point 8: Line 185 - how can you have almost 41 females?

Response 8: The participants are recruited randomly from PCUs, and apart from this, female involvement in the healthcare system is much higher than that of their male counterparts globally, and this population that was studied also experienced this fact.

Point 9: The actual interview section was interesting, although 8 is a very small number. I am not sure what conclusions one can make, except the remarks you include seem to be very predictable. What I would be more interested in would be the differences in the remarks. What comments were the same and how many and what and how many were different?

Response 9: A quantitative survey was used to collect data to know how many participants responded to a particular question. While, this aspect (qualitative method) the reviewer is talking about is to comprehend the life experiences faced by the participants, this part does not intend to count similar responses but to capture relevant information that the research questions do not foresee. This is also one of the reasons mentioned above why the questions asked by the participants are open-ended and general. From the literature, the number of participants recommended for a phenomenological study is between 5 – 50, which is where the number recruited (8) in this study falls within the range.

This article “Dworkin, S.L. Sample Size Policy for Qualitative Studies Using In-Depth Interviews. Arch Sex Behav 41, 1319–1320 (2012). https://doi.org/10.1007/s10508-012-0016-6” give more detail about our responses here.

Point 10: Finally, I find the conclusions very weak.  What were you trying to accomplish by writing this paper?  Was the purpose to get change?  Was it to identify problems that should be corrected?  There is little evaluation of what is good and what is less good. The Conclusions should be the strongest part of the paper.  What did you intend to be of value to the reader, including readers from another country?

Response 10: Done

Round 2

Reviewer 1 Report

The authors have addressed all concerns and issues raised.